# Comparison of the Amount of Used and the Ease of Oral Care between Liquid and Gel-Type Oral Moisturizers Used with an Oral Care Simulators

**DOI:** 10.3390/ijerph19138158

**Published:** 2022-07-03

**Authors:** Hiroyuki Suzuki, Junichi Furuya, Chiaki Matsubara, Michiyo Aoyagi, Maki Shirobe, Yuji Sato, Haruka Tohara, Shunsuke Minakuchi

**Affiliations:** 1Gerodontology and Oral Rehabilitation, Department of Gerontology and Gerodontology, Graduate School of Medical and Dental Sciences, Tokyo Medical and Dental University, 1-5-45 Yushima, Bunkyo-ku, Tokyo 113-8549, Japan; h.suzuki.gerd@tmd.ac.jp (H.S.); s.minakuchi.gerd@tmd.ac.jp (S.M.); 2Department of Geriatric Dentistry, Showa University School of Dentistry, 2-1-1 Kitasenzoku, Ohta-ku, Tokyo 145-8515, Japan; sato-@dent.showa-u.ac.jp; 3Dysphagia Rehabilitation, Department of Gerontology and Gerodontology, Graduate School of Medical and Dental Sciences, Tokyo Medical and Dental University, 1-5-45 Yushima, Bunkyo-ku, Tokyo 113-8549, Japan; sacra333march@gmail.com (M.A.); haruka-t@rd5.so-net.ne.jp (H.T.); 4Department of Dental Hygiene, University of Shizuoka, Junior College, 2-2-1 Oshika, Suruga-ku, Shizuoka 422-8021, Japan; m.chiakingyo@gmail.com; 5Research Team for Promoting Independence and Mental Health, Tokyo Metropolitan Institute of Gerontology, 35-2 Sakae-cho, Itabashi-ku, Tokyo 173-0015, Japan; mashirobe@gmail.com

**Keywords:** oral care, oral moisturizers, dysphagia, aspiration pneumonia, multidisciplinary medical care

## Abstract

Oral care involving the removal of dry sputum is effective for older patients who require nursing care or hospitalization. However, safe and efficient oral care methods for such patients remain unclear. We aimed to simulate the oral cavity of older adults with dry mouth and elucidate the differences between two moisturization agents, water and gel-like oral moisturizer, and investigate the effect of occupation and experience on the amount of use and the ease of oral care. Using an oral care simulator (MANABOT^®^, Nissin Dental Products Inc., Kyoto, Japan), 42 students and 48 dental professionals (13 dentists and 35 dental hygienists) performed oral care using moisturization agents to facilitate dry sputum removal. The time required for oral care, amount of water or gel used, amount of pharyngeal inflow, and ease of oral care when using water or gel were compared. The simulations revealed that the amount of use and pharyngeal inflow for gel (2.9 ± 1.6 and 0.3 ± 0.3, respectively) were significantly lower than those for water (6.8 ± 4.1 and 1.2 ± 1.5, respectively) in all participants. Using a gel-like moisturizer might reduce the aspiration risk in older patients requiring nursing care or hospitalization, regardless of occupation and experience.

## 1. Introduction

Oral functions, such as swallowing and oral hygiene, decline with age [1,2]. This decline is associated with a decrease in nutritional status [3] and increased mortality in older adults [4,5,6,7]. Dysphagia, which is caused by various diseases, such as stroke [8] or dementia [9], is one of the most frequently observed oral dysfunctions in older adults requiring long-term nursing care and hospitalization [10]. Poor oral hygiene and dysphagia are significantly associated with the occurrence of aspiration pneumonia, which results from the introduction of oral bacteria into the respiratory tract [11,12]. As aspiration pneumonia increases the hospitalization duration and mortality rate, it is important to prevent it in older patients [13,14]. Therefore, oral hygiene management of hospitalized older adults or older patients requiring nursing care is important to prevent aspiration pneumonia.

Dry mouths are often observed in hospitalized older adults or older adults requiring long-term care [15,16]. Because of a decrease in the self-cleansing effect caused by dry mouth [17], food debris, exfoliated mucous membranes, and sputum may accumulate in the oral cavity. In cases of dry mouth, these stains, such as dried sputum, stick to the tongue and palate and become a risk factor for halitosis and aspiration pneumonia [11,18]. Forcibly removing dry sputum causes bleeding and pain; therefore, it is necessary to moisturize and then remove dry sputum during oral care [19]. Oral moisturizers, such as water or gel, are used to moisten the dry sputum in the oral cavity. As water is highly fluid, it tends to moisten dry sputum well; however, it tends to flow into the pharynx, thus, increasing the risk of aspiration. In contrast, oral moisturizers require time to moisturize dry sputum and are retained in the oral cavity. If the oral cavity is very dry, oral moisturizers tend to be retained in layers over the dry sputum on the mucosa. The significance of oral management through multidisciplinary collaboration for hospitalized older adults and older adults requiring long-term nursing care has been clarified in recent literature [20,21,22,23]. In fact, oral care for these patients is often provided by nurses and caregivers, in addition to dental professionals. Therefore, it is necessary to develop standardized oral care methods that can be implemented safely and efficiently without relying on specialized knowledge and experience. Further, the aspiration risk associated with the liquid or oral moisturizers is the greatest challenge in this regard; however, measuring the pharyngeal inflow of liquid and oral moisturizers in a living body is difficult. Moreover, little has been clarified regarding the efficient oral care methods, including the ease of oral care and the time required for oral care when using liquids and oral moisturizers.

Poor oral hygiene and dry sputum are often observed in hospitalized older adults and older adults requiring long-term nursing care. Therefore, in this study, we performed simulations using an oral care robot to compare the amount of use and the ease of oral care between liquid and gel-type oral moisturizers for these conditions. We also compared the amount of moisturization agents used and ease of oral care between students and dental professionals to determine the effect of oral care experience for each moisturizer.

## 2. Materials and Methods

### 2.1. Participants

The participants included dentists and dental hygienists working at the Tokyo Medical and Dental University School of Dentistry and students training to become dental hygienists at the Department of Oral Health Sciences, Tokyo Medical and Dental University School of Dentistry. We explained the study to all participants in writing and verbally and obtained informed consent before initiating the study. This study was approved by the Institutional Review Board of the Faculty of Dentistry, Tokyo Medical and Dental University (approval number D2016-096) and was carried out in accordance with the tenets of the Declaration of Helsinki.

### 2.2. Study Protocol

In this study, oral care simulations were performed using an oral care simulator (MANABOT^®^, Nissin Dental Products Inc., Kyoto, Japan) that replicated the oral cavity of hospitalized older adults and older adults requiring long-term nursing care. Dry mouth, which is common among these patients, was present in the simulator’s oral cavity. Assuming that the patient cannot maintain oral care independently, pseudo-plaque was added to the maxillary palate and the lingual side of the lower mandibular molars, as shown in Figure 1. The pseudo-plaque was prepared by mixing 1 g of thickening agent (Toromi Up Perfect, Nisshin Oillio, Tokyo, Japan) and 2 mL of water in advance. It was added at two sites in the oral cavity (maxillary palate, 0.9 g; mandibular left molar lingual side, 0.1 g) and dried at room temperature (23 °C) for 8 h. Moreover, the pseudo-plaque was colored green using food coloring for easier identification by study participants. The posture of the simulator was set to a reclining position of 30°, and oral care was simulated by all participants under the following two conditions: (1) when using water as an oral moisturizer and (2) when using a gel-like oral moisturizer (Okuchi wo arau gel, Nippon Shika Yakuhin Co., Ltd., Shimonoseki, Japan). The main components of this gel-like oral moisturizer are water, glycerin, hydroxyethyl cellulose, sodium polyacrylate, and sodium benzoate. A sponge brush, a toothbrush, an oral care gauze, an aspirator with a suction tube, and water or a gel-like oral moisturizer (hereinafter referred to as gel) were used. Oral care was performed in a series of steps, as follows [24]: (1) the oral cavity was moisturized with an oral moisturizer and sponge brush; (2) pseudo-plaque was removed with a toothbrush; (3) dry sputum was collected with a sponge brush. This procedure was repeated to remove the pseudo-plaque. Prior to study initiation, the oral care method was fully explained to participants through a cohesive and comparable presentation using slides and articles. The order of the oral care simulations (first use of water or gel) was randomly determined using the Latin square design for each participant, and the order was determined by a dentist not directly involved in the experiments or outcome assessments. Oral care was terminated when the entire pseudo-plaque was removed.

### 2.3. Outcomes

The time required for oral care, amount of water or gel used, and amount of pharyngeal inflow were measured as outcomes. The amount of pharyngeal inflow was calculated by placing dry gauze on the pharynx of the simulator and measuring the weight before and after oral care. The oral care simulator used in this study has a structure that does not allow water to be stored in the simulator’s mouth. Therefore, as an alternative method, we chose to measure pharyngeal inflow by measuring the weight change of dried gauze placed on the pharynx of the simulator before and after oral care. In addition, a self-administered subjective evaluation of the ease of oral care using water and gel was performed using a 100-mm visual analog scale (VAS), and the participants were asked to evaluate the following: “ease of moisturization of dry sputum,” “ease of dry sputum removal,” “ease of dry sputum collection,” and “overall evaluation of ease of oral care.” “Overall evaluation of ease of oral care” was an evaluation of the ease of oral care when judging the wettability, removability, and retrievability of each oral moisturizer used as a whole. In this 100-mm VAS, scores of 0 and 100 points indicated not easy and very easy procedures, respectively. We also surveyed dental professionals using a questionnaire, in which the number of years of dental care experience was recorded. In addition, participants were asked to rate the frequency of oral care for older adults requiring long-term nursing care in daily clinical practice (“never,” “rarely,” “sometimes,” “often,” and “daily”). Those who answered “never” were considered to have no experience in managing oral care for older adults requiring long-term nursing care, while those who answered otherwise were considered to have experience. All outcome evaluations were conducted by two dental hygienists who were fully trained in the evaluation criteria.

### 2.4. Sample Size Estimation

The sample size was calculated from preliminary experiments with dentists before this study, assuming a pharyngeal inflow rate of 1.0 g (SD: 1.2) when water was used and 0.5 g (SD: 0.5) when gel-like moisturizer was used. Thirty-eight participants were required for an 80% power, with a two-sided alpha level of 0.05. A minimum sample size of 76 students and dental professionals was required for this study since we planned to compare the use of water and gel-like moisturizers between them.

### 2.5. Statistical Analysis

Statistical analysis was performed on the required time for oral care, amount of water or gel used, amount of pharyngeal inflow, and ease of oral care (ease of moisturization, ease of removal, ease of collection, and overall evaluation of ease of oral care), which were the outcomes of this study. In addition, the outcomes between water and gel-like moisturizer were compared using the Wilcoxon signed-rank test for all participants and among dental professionals or students, respectively. Moreover, comparisons based on the moisturizer used were performed for all study participants. Furthermore, the Mann–Whitney U test was used to compare outcomes based on the participants’ characteristics (whether they are dental professionals or students) and to compare outcomes based on oral care experience among dental professionals for older adults requiring long-term nursing care. SPSS Statistics Ver. 27 (IBM Corp., Armonk, NY, USA) was used for statistical analysis, and the significance level was set to 5%.

## 3. Results

### 3.1. Participant Characteristics

The participants in this study comprised 42 oral health science students and 48 dental professionals (13 dentists and 35 dental hygienists). The students had no previous experience in oral care. The average number of years of oral care experience for dental professionals was 7.0 ± 7.9 years; 19 (39.6%) dental professionals had no experience in oral care for older adults requiring long-term nursing care, while the remaining 29 (60.4%) had some experience.

### 3.2. Comparison by Type of Oral Moisturizer Used

Table 1 shows a comparison of outcomes when oral care was performed with water vs. when it was performed with gel by all participants in this study. Compared to oral care with water, oral care with gel required significantly more time, although the amount used and the pharyngeal inflow were significantly less. Furthermore, the “ease of collection” was also significantly greater. Table 2 and Table 3 show a comparison of outcomes when oral care was performed with water and gel by dental professionals and students, respectively. In both groups, there were no significant differences between oral care using water and oral care using gel in terms of the ease of oral care. Conversely, when performing oral care using gel, the amount used and the pharyngeal inflow were significantly less in both groups than that observed in oral care using water. In the student group, oral care using gel took longer than oral care using water.

### 3.3. Comparison by Differences in Research Participant Characteristics

Table 4 compares outcomes when oral care was performed with water and gel, each by dental professionals and students. No significant difference in outcomes was observed between dental professionals and students for most items with either of the oral moisturizers. However, pharyngeal inflow in oral care using water was significantly higher in students than in dental professionals.

### 3.4. Comparison of Dental Professionals Based on Their Experience in Oral Care for Older Adults Requiring Long-Term Nursing Care

Table 5 compares outcomes when oral care using water and gel was performed in the group with and without oral care experience for older adults requiring long-term nursing care among dental professionals. For oral care using water, no significant difference in outcomes was observed based on the presence or absence of oral care experience. However, for oral care using gel, the group with oral care experience had significantly more pharyngeal inflow than the group without oral care experience, although the “ease of collection” was significantly greater.

## 4. Discussion

In this study, students and dental professionals performed oral care using water or gel as an oral moisturizer for dry sputum on an oral care simulator. The results revealed that the amount of oral moisturizer used and the pharyngeal inflow was small when oral care was performed using gel. Furthermore, in comparing the pharyngeal inflows generated during oral care by students and dental professionals, there was no significant difference between the two groups when the gel was used. However, when water was used as an oral moisturizer, the pharyngeal inflow was significantly greater for students than dental professionals. These are the results of a simulation model in which the mucous membrane did not absorb water and moisturizer. However, the results suggest that since only a small amount of gel-like oral moisturizer is required, the risk of aspiration during oral care is reduced, and, therefore, the gel is useful as an effective oral care method even for dental professionals. Moreover, we believe oral care using a gel-like moisturizer may reduce the risk of aspiration even without specialized skill or experience.

Liquids are often used to moisturize and remove contaminants during oral care [25,26]. However, while liquids tend to moisturize dry sputum, they also tend to flow into the pharynx, which may increase the aspiration risk, especially during oral care for older adults with dysphagia. Ikeda et al. reported that oral bacteria could be significantly reduced by wiping off debris with a wet wipe after oral care with water and a gel-like oral moisturizer [27]. From this report, it can be stated that oral care using a form of moisturizer other than water is also effective. As the gel-like oral moisturizer has an appropriate viscosity, it may be effective for oral care of patients at high risk of aspiration. In the present study, in the student group, an average of approximately 2 mL of water remained in the oral cavity and pharynx when oral care was performed using only liquids. Our findings show that care must be taken when using liquids for oral care, especially if the person providing the oral care lacks experience, suggesting the importance of the moisturizing method during oral care. Oral care using a gel-like moisturizer was sufficiently efficient for dental professionals and those with little skill or experience. In contrast, among the dental professionals in this study, there was a significant difference in the “ease of collection” when using a gel-like oral moisturizer, depending on the level of oral care experience.

It has been reported that gel-like oral moisturizers are effective not only from the viewpoint of pharyngeal inflow but also in improving dry mouth symptoms and dry mouth [28,29]. Furuya et al. found that among 459 patients admitted to acute care hospitals with dysphagia, approximately 70% had dry mouth, and approximately half had moderate or severe dry mouth [30]. It has also been reported that more than 50% of patients targeted for nutrition support teams in acute care hospitals and patients with terminal cancer who were targeted for palliative care had dry mouth problems [21,23]. It has been reported that dry mouth was observed in more than 50% of patients admitted to chronic hospitals [31]. Further, Yoon et al. reported that of 559 randomly selected older adults requiring long-term nursing care admitted to a facility, who completed a survey on the oral environment, approximately 40% had dry mouth [32]. In addition, it has been reported that the number of bacteria and fungi in the oral cavity increases in patients with severe dry mouth [33,34]. A previous study reported that oral care could effectively reduce the number of bacteria in the oral cavity by combining a mouthwash and oral moisturizer [34]. Considering this, gel-like oral moisturizers may be highly useful in actual clinical practice, especially in difficult cases with severe xerostomia and staining; therefore, efficient oral care may be achieved using a gel-like oral moisturizer.

This study had several limitations. First, the pseudo-plaque used in this study was limited to only two sites (i.e., the maxillary palate and the lingual side of the left mandibular molar), and it was only verified by the simulator. The maxillary palate and the lingual side of the left mandibular molar are relatively easy to see directly. In addition, as the verification was performed using a simulator, unlike in actual patients, the mouth was always open, which made oral care relatively easy. In the actual clinical setting, patients may have mouth-opening difficulties, inability to open the mouth, involuntary movements, and layers of stains in difficult-to-see areas. Moreover, as this study was conducted using a simulator, it is impossible to reproduce surface adhesion, absorption, wettability, and plaque retention of dry phlegm that are entirely consistent with those of the oral cavity. In the future, it is necessary to conduct intervention studies to verify whether gel-like oral moisturizers are effective for such cases. Second, the time required for oral care in this study was likely longer than that in actual clinical practice because the time required to remove all pseudo-plaque was evaluated as the time required for oral care in this study. When oral care is provided in hospitals or facilities by nurses or caregivers, the time available for oral care is often limited because of various constraints. In contrast, oral care by dental professionals in hospitals or facilities is often provided on request, allowing more time to be spent on it than when it is performed by nurses or caregivers. Therefore, our findings might reflect the situation when oral care is provided by dental professionals. Third, in this study, we examined the differences in the presence or absence of oral care experience for older adults requiring long-term nursing care among dental professionals. However, the level of the general condition of patients undergoing oral care by dental professionals has not been examined, and the oral care skills of these professionals were not examined. Finally, students were included in this study because they were assumed to have little dental expertise and no clinical experience. However, the students who participated in this study were undergoing training to become dental hygienists and were considered to have some expertise compared to nurses and caregivers. Considering that nurses and caregivers who do not have specialized knowledge also administer oral care in actual clinical settings, similar verification should be performed in such occupations.

## 5. Conclusions

Considering the above limitations, it can be concluded that using gel-type oral moisturizer results in a significantly smaller residual amount in the pharynx than when liquids were used in oral care. Therefore, we believe that due circumspection is necessary when using liquids in oral care if the person providing the oral care has limited skills. These results suggest that oral care using a gel-like oral moisturizer may be effective when performing oral care for patients with dysphagia.

## Figures and Tables

**Figure 1 ijerph-19-08158-f001:**
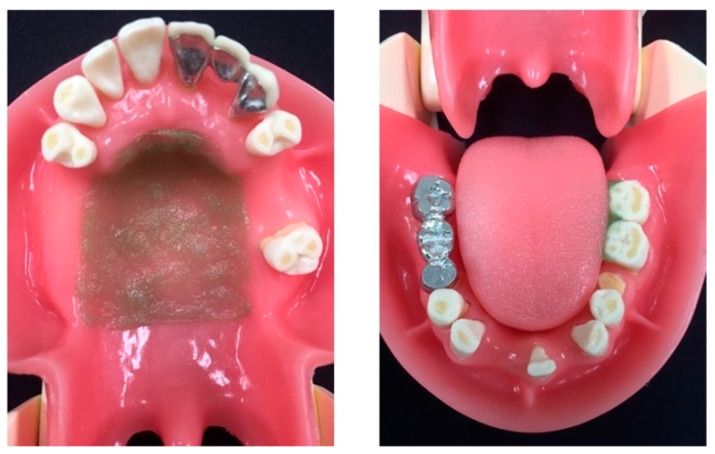
Pseudo-plaque attached to the simulator.

**Table 1 ijerph-19-08158-t001:** Comparison of oral care with water and gel among all participants (*n* = 90).

		Water		Gel		*p*-Value
		Mean ± SD	Median	Mean ± SD	Median	
Time required	(min)	9.1 ± 2.5	8.5	10.0 ± 3.3	9.7	0.007 *
Amount used	(g)	6.8 ± 4.1	6.0	2.9 ± 1.6	2.7	<0.001 *
Amount of pharyngeal inflow	(g)	1.2 ± 1.5	0.6	0.3 ± 0.3	0.2	<0.001 *
Ease of moisturization	(points)	57.6 ± 26.4	62.0	62.3 ± 19.4	64.0	0.150
Ease of removal	(points)	59.3 ± 21.2	63.0	58.2 ± 18.5	61.0	0.764
Ease of collection	(points)	52.1 ± 22.6	52.0	58.1 ± 20.5	60.0	0.041 *
Overall evaluation of ease of oral care	(points)	59.8 ± 19.7	67.0	59.9 ± 17.5	61.0	0.874

* *p* < 0.05 Wilcoxon signed rank test; SD, standard deviation.

**Table 2 ijerph-19-08158-t002:** Comparison of oral care with water and gel among dental professionals (*n* = 48).

		Water		Gel		*p*-Value
		Mean ± SD	Median	Mean ± SD	Median	
Time required	(min)	9.0 ± 2.1	8.5	9.7 ± 3.0	9.4	0.169
Amount used	(g)	6.4 ± 4.6	5.4	2.8 ± 1.6	2.3	<0.001 *
Amount of pharyngeal inflow	(g)	0.8 ± 1.0	0.4	0.2 ± 0.2	0.1	<0.001 *
Ease of moisturization	(points)	57.5 ± 25.9	66.0	64.7 ± 18.7	64.0	0.287
Ease of removal	(points)	61.6 ± 19.1	65.0	59.0 ± 18.4	61.0	0.462
Ease of collection	(points)	55.6 ± 21.1	54.0	59.8 ± 19.5	61.0	0.385
Overall evaluation of ease of oral care	(points)	62.9 ± 18.4	67.5	62.2 ± 15.9	63.0	0.608

* *p* < 0.05 Wilcoxon signed rank test; SD, standard deviation.

**Table 3 ijerph-19-08158-t003:** Comparison of oral care using water and gel among students (*n* = 42).

		Water		Gel		*p*-Value
		Mean ± SD	Median	Mean ± SD	Median	
Time required	(min)	9.1 ± 2.9	8.7	10.1 ± 3.4	10	0.016 *
Amount used	(g)	7.2 ± 3.4	7.3	3.1 ± 1.6	2.9	<0.001 *
Amount of pharyngeal inflow	(g)	1.7 ± 1.8	1.3	0.3 ± 0.3	0.2	<0.001 *
Ease of moisturization	(points)	57.7 ± 22.5	56.5	60.0 ± 20.0	62.5	0.291
Ease of removal	(points)	56.7 ± 23.9	60.0	57.3 ± 18.9	59.5	0.804
Ease of collection	(points)	48.2 ± 23.9	49.5	56.2 ± 21.7	58.5	0.081
Overall evaluation of ease of oral care	(points)	56.5 ± 20.8	64.0	57.3 ± 19.1	60.0	0.756

* *p* < 0.05 Wilcoxon signed rank test; SD, standard deviation.

**Table 4 ijerph-19-08158-t004:** Comparison of oral care in each group of dental professionals and students.

			Dental Professionals (*n* = 48)		Oral Health Science Students (*n* = 42)		*p*-Value
			Mean ± SD	Median	Mean ± SD	Median	
**Water**	Time required	(min)	9.0 ± 2.1	8.5	9.1 ± 2.9	8.7	0.777
	Amount used	(g)	6.4 ± 4.6	5.4	7.2 ± 3.4	7.3	0.094
	Amount of pharyngeal inflow	(g)	0.8 ± 1.0	0.4	1.7 ± 1.8	1.3	0.008 **
	Ease of moisturization	(points)	57.5 ± 25.9	66.0	57.7 ± 22.5	56.5	0.840
	Ease of removal	(points)	61.6 ± 19.1	65.0	56.7 ± 23.9	60.0	0.326
	Ease of collection	(points)	55.6 ± 21.1	54.0	48.2 ± 23.9	49.5	0.190
	Overall evaluation of ease of oral care	(points)	62.9 ± 18.4	68.0	56.5 ± 20.8	64.0	0.225
**Gel**	Time required	(min)	9.7 ± 3.0	9.4	10.1 ± 3.4	10.0	0.574
	Amount used	(g)	2.8 ± 1.6	2.3	3.1 ± 1.6	2.9	0.224
	Amount of pharyngeal inflow	(g)	0.2 ± 0.2	0.1	0.3 ± 0.3	0.2	0.083
	Ease of moisturization	(points)	64.7 ± 18.7	64.0	60.0 ± 20.0	62.5	0.289
	Ease of removal	(points)	59.0 ± 18.4	61.0	57.3 ± 18.9	59.5	0.557
	Ease of collection	(points)	59.8 ± 19.5	61.0	56.2 ± 21.7	58.5	0.363
	Overall evaluation of ease of oral care	(points)	62.2 ± 15.9	63.0	57.3 ± 19.1	60.0	0.269

** *p* < 0.05 Mann–Whitney U test; SD, standard deviation.

**Table 5 ijerph-19-08158-t005:** Comparison of oral care with or without oral care experience for older adults in dental professionals.

			Without Oral Care Experience (*n* = 19)		With Oral Care Experience (*n* = 29)		*p*-Value
			Mean ± SD	Median	Mean ± SD	Median	
**Water**	Time required	(min)	9.2 ± 1.8	9.3	8.9 ± 2.3	8.2	0.370
	Amount used	(g)	6.4 ± 3.6	5.8	6.5 ± 5.3	5.1	0.697
	Amount of pharyngeal inflow	(g)	0.9 ± 1.1	0.5	0.7 ± 0.9	0.4	0.370
	Ease of moisturization	(points)	55.4 ± 23.5	65.0	58.8 ± 27.7	67.0	0.405
	Ease of removal	(points)	60.4 ± 19.4	65.0	62.4 ± 19.2	61.0	0.841
	Ease of collection	(points)	57.6 ± 20.6	51.0	54.2 ± 21.7	55.0	0.576
	Overall evaluation of ease of oral care	(points)	65.8 ± 12.4	69.0	60.9 ± 21.4	65.0	0.555
**Gel**	Time required	(min)	10.1 ± 2.7	9.6	9.5 ± 3.3	9.2	0.250
	Amount used	(g)	2.6 ± 1.8	2.0	2.7 ± 1.5	3.1	0.382
	Amount of pharyngeal inflow	(g)	0.2 ± 0.3	0.1	0.2 ± 0.2	0.2	0.030 **
	Ease of moisturization	(points)	59.7 ± 18.3	60.0	68.0 ± 18.5	68.0	0.140
	Ease of removal	(points)	53.2 ± 22.1	61.0	62.8 ± 14.6	67.0	0.129
	Ease of collection	(points)	51.6 ± 18.7	53.0	65.2 ± 18.4	70.0	0.010 **
	Overall evaluation of ease of oral care	(points)	58.8 ± 16.4	59.0	64.4 ± 15.4	67.0	0.246

** *p* < 0.05 Mann–Whitney U test; SD, standard deviation.

## Data Availability

The datasets analyzed in this study are available from the corresponding author on reasonable request.

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
