# Peer review of "Comparison of the Amount of Used and the Ease of Oral Care between Liquid and Gel-Type Oral Moisturizers Used with an Oral Care Simulators"

_ijerph, 2022, doi:10.3390/ijerph19138158_

Round 1

Reviewer 1 Report

To Authors

  • Title

The title expresses clearly what the manuscript is about. But, it could be excessive to refer to “efficiency and safety” in such reproducible experimental conditions based on oral care simulator, especially in relation to the results of the study.

- Abstract

The abstract clearly summarizes the paper with enough information to stand alone. Revise lines 28-29 and specify with respect to what was used less oral moisturizer.

  • Introduction

The introduction summarizes the current state of the topic, but the limitations of current knowledge should be better described, explaining why the study was conducted. This section should explain what is meant to be the “dry sputum” clearly for a better comprehension of the reader.   

  • Methods

The study design and methods for the research question present several limitations as explained, confining the results only to an experimental context; more appropriate results could be obtained from clinical trials. In fact, the use of an oral care simulator does not make comparable the results that could be obtained from clinical conditions. The same consideration should be made for the pseudo-plaque created with a thickening agent and oral care practice. The oral conditions of dryness will never be like those of a simulator (e.g in terms of adhesion, absorption, wettability of surfaces and plaque retention).

Doubts for this section:

- Why calculate pharyngeal inflow by evaluating the weight of the gauze rather than by the direct weight of the simulator’s oropharyngeal reflux fluids? Explain.

- Although the VAS can be used for a subjective evaluation of the variables, it would be more appropriate to include in the text a rating scale with a minimal description, especially for the "total score of dry sputum”.

  • Results

The results are presented clearly and accurately by measures and tools used to evaluate outcomes variables, closely matching the methods. The data described in the text are consistent with those in the tables. Unfortunately, given these limitations, the scientific validity of the results obtained is rather limited to the predictable.

  • Discussion and conclusion

In the discussion section, findings are logically explained by main topics, comparing the results with current outcomes in the research field. Also presented are the strengths points and limitations of the study. The implications of the findings for future research and potential applications are discussed.

  • Tables and figures

Data are presented in a clear and appropriate manner, with a presentation of tables that is consistent with the text. 

Reviewer 2 Report

I congratulate the authors for the manuscript.

I suggest to expand the Introduction explaning the connection between this problem and the halitosis, including the valuable work by Prof Dolci et al. (Silvia D'Agostino et al. 2021, Halitosis awareness among italian dentists, hygienists and students. Int J Dent &Ora Hea. 7:2 ).

Reviewer 3 Report

The manuscript is not well written and clear. Findings
  • The Methods are confusing
  • Does not adhere to the PICO format.
  • The dependent variable is not clearly.

Round 2

Reviewer 3 Report

The Methods are not clear.   Add to the manuscript more details about the interventions.  I think readers would benefit from seeing some of those details in this text. Randomization process is lacking details. Who conducted the randomization process?  Variables. Why compare students and professionals? How did you calculate the sample size?
